# A Study on Visible Light Communication Systems Based on OLED Devices

**DOI:** 10.3390/mi16121338

**Published:** 2025-11-27

**Authors:** Wanyi Zhang, Haochen Xu, Sihang Ji, Jiazhuang Lan

**Affiliations:** 1College of Information Technology, Jilin Engineering Research Center of Optoelectronic Materials and Devices, Jilin Normal University, Siping 136000, China; 2Jilin Provincial Key Laboratory of Wide Bandgap Semiconductor Material Growth and Device Applications, Jilin Normal University, Changchun 130103, China

**Keywords:** organic light-emitting diodes (OLEDs), modulation bandwidth, visible light communication, wavelength division multiplexing

## Abstract

Addressing the limitations of conventional inorganic light-emitting diodes (LEDs) in flexible visible light communication (VLC) applications, this study investigates the feasibility of organic light-emitting diodes (OLEDs) as an integrated platform for illumination, display, and communication. The optoelectronic characteristics and modulation bandwidth of red, green, and blue (RGB) OLEDs were systematically measured. Based on the experimental data, a wavelength division multiplexing (WDM) VLC system employing non-return-to-zero on-off keying (NRZ-OOK) modulation was constructed in simulation software for validation. The results indicate stable optoelectronic performance for all three primary-color OLEDs, with a maximum modulation bandwidth of 466 kHz achieved for the blue device. The system simulation demonstrates stable parallel transmission of three independent data channels, attaining a minimum bit error rate (BER) as low as
$3.74\times 10^{-35}$ achieved for the green device. This work confirms the potential of OLEDs for emerging communication applications such as flexible displays and wearable devices.

## 1. Introduction

As research on sixth-generation (6G) mobile communication technology gradually unfolds, the vision of an intelligent interconnection of all things places unprecedented demands on the bandwidth, connection density, and security of wireless communications. Although conventional radio frequency (RF) communication technologies have matured, they increasingly face challenges such as spectrum scarcity, susceptibility to electromagnetic interference, and information security risks. In this context, Visible Light Communication (VLC)—a wireless communication technology that employs the 380–780 nm visible light band as the information carrier—is widely regarded as a crucial complementary technology in the 6G era [1]. It offers distinctive advantages including license-free operation, absence of electromagnetic radiation, high confidentiality, and inherent compatibility with illumination functions. The application scope of VLC is rapidly expanding from traditional fields such as indoor positioning and Li-Fi to strategic emerging domains like the Internet of Vehicles (V2X), the Industrial Internet of Things (IIoT), and integrated air–ground–sea–space information networks [2], demonstrating substantial potential for future development.

However, the core light-emitting components in current VLC systems are predominantly based on inorganic light-emitting diodes (LEDs) or laser diodes (LDs). Conventional LEDs, which function as rigid point sources, are characterized by fixed emission patterns and limited spectral tunability. These inherent limitations hinder their compatibility with next-generation human-machine interfaces—such as flexible electronics, wearable devices, and curved displays—which demand device lightweightness and form-factor freedom [3]. Consequently, the deep integration and innovative application of VLC technology in broader scenarios have been somewhat restricted. The emergence of organic light-emitting diodes (OLEDs) presents a groundbreaking solution to this bottleneck. OLEDs possess a unique set of properties, including large-area surface emission, soft and diffuse light, a spectrum that can be precisely “tailored” at the molecular level, and intrinsic flexibility. These characteristics make OLEDs ideally suited for the evolving trends of “display-communication integration” and “illumination-communication integration.” As such, OLEDs can not only function as high-quality light sources for displays and lighting but also naturally serve as distributed optical antennas, thereby laying a physical foundation for truly “ubiquitous optical connectivity.”

In recent years, both academia and industry have engaged in a series of cutting-edge explorations of OLED-based Visible Light Communication (OLED-VLC), achieving remarkable breakthroughs in enhancing its communication performance. In material and device structure optimization, a joint team from the University of St Andrews and the University of Cambridge precisely selected dibenzopyrene (DNP)—a stable organic compound—as the emissive material and systematically optimized the OLED layer structure and thickness. This approach yielded a record data transmission performance, achieving rates of 4.0 Gbps at 2 m and 2.9 Gbps at 10 m, thereby setting a new benchmark in the OLED-VLC field and demonstrating its feasibility for indoor scenarios [4]. In system integration, a joint team from the Hon Hai Research Institute successfully developed a multi-wavelength micro-LED (μ-LED) system incorporating red, yellow, green, and blue emitters. By employing wavelength division multiplexing (WDM) technology, they significantly increased the system’s transmission capacity, providing a critical reference for the future integration of OLEDs into WDM-VLC systems [5]. Regarding signal processing and equalization techniques, Min et al. proposed a novel pre-equalizer construction method based on vector-fitting-optimized filter synthesis. By developing an active-passive hybrid equalization circuit, they extended the 3-dB bandwidth of a commercial phosphorescent white LED system from 30 MHz to 600 MHz. Although this study focused on LEDs, its equalization strategy offers significant reference value for mitigating the bandwidth limitations of OLEDs [6]. In the domain of system security, Sun et al. investigated multi-user physical layer security in intelligent reflecting surface (IRS)-assisted VLC systems. They proposed an optimization scheme based on an iterative Kuhn-Munkres algorithm. By optimizing the allocation of IRS units, they substantially improved the secrecy performance of the VLC system, providing a new technical avenue for addressing security concerns in future OLED-VLC systems [7]. Collectively, these research advancements propel OLED-VLC technology toward practical implementation from multiple critical dimensions.

Nevertheless, the integration of OLEDs into VLC systems confronts significant challenges. The primary bottleneck is their inherently limited modulation bandwidth, which is substantially inferior to that of conventional LEDs and LDs. This limitation originates from fundamental physical mechanisms in organic semiconductors, including relatively low charge carrier mobility, high parasitic capacitance in device architectures, and long excited-state exciton lifetimes [8]. The constrained bandwidth severely restricts the achievable data transmission rates in OLED-VLC systems, which constitutes a major technical obstacle that must be surmounted for high-speed communication. Furthermore, critical aspects of VLC systems employing RGB primary-color OLEDs remain insufficiently explored and require in-depth investigation. These include the inherent bandwidth variation among different color-emitting devices—where blue OLEDs typically exhibit greater bandwidth potential than their red counterparts—as well as system integration schemes and a comprehensive evaluation of the overall communication performance.

## 2. Characterization of OLED Devices

This study employed high-efficiency, top-emitting red, green, and blue (RGB) organic light-emitting diodes (OLEDs) devices, which were provided by PhiChem Corporation. These devices, fabricated via a vacuum thermal evaporation process, feature a 2 mm × 2 mm light-emitting area and a 25 mm × 25 mm substrate. A schematic diagram of the OLED structure is presented in Figure 1. The active emitting area is represented by the four small yellow squares at the center. The devices were powered by a linear regulated voltage source. The anode and cathode were connected using red and black alligator clips, respectively. Illumination was achieved once the supplied voltage exceeded the turn-on voltage of the OLED. As shown in Figure 2, the devices are encapsulated in glass, with a substrate structure of indium tin oxide (ITO)/Ag/ITO (with thicknesses of 10 nm/100 nm/10 nm). Compared to bottom-emitting OLEDs, the top-emitting architecture employed here offers distinct advantages for VLC systems, including higher light extraction efficiency, microcavity-enhanced signal quality, superior high-speed modulation capability, and greater potential for integration into diverse scenarios.

Figure 3 depicts the energy level diagrams of the RGB OLED devices. The energy level data were provided by PhiChem Corporation. These diagrams systematically elucidate the energy level alignment mechanism across the entire functional stack in each device. A consistent device architecture employs ITO as the anode, with a highest occupied molecular orbital (HOMO) energy level of approximately −4.70 eV, combined with a hole transport layer (HTL) composed of 3,4-ethylenedioxythiophene oxocarbenium ion (COO::EDOT^+^) to form the injection interface. The HOMO level of the HTL was modulated by ultraviolet (UV) irradiation. After 10 min of exposure, the HOMO level reached −4.88 eV, resulting in a hole injection barrier of 0.18 eV at the interface with ITO. Concurrently, this establishes a favorable energy level step with the underlying Tris(4-carbazoyl-9-ylphenyl)amine (TCTA) layer (HOMO = −5.60 eV), thereby promoting efficient and directional hole transport.

The intermediate layer utilizes *N*,*N*′*-Bis*(naphthalen-1-yl)-*N*,*N*′*-bis*(phenyl)benzidine (NPB) (HOMO = −5.40 eV, Lowest Unoccupied Molecular Orbital (LUMO) = −2.40 eV) as a transitional functional material. TCTA, with its lower-lying energy levels (HOMO = −5.60 eV, LUMO = −2.20 eV), effectively facilitates hole acceptance and suppresses electron leakage. The emissive layer (EML) employs 4,4′-Bis(*N*-carbazolyl)-1,1′-biphenyl (CBP) as the host material (HOMO = −6.00 eV, LUMO = −2.90 eV), which is respectively doped with the red-emitting phosphor Ir(pq)_2_acac (HOMO = −5.20 eV, LUMO = −3.00 eV), the blue-emitting phosphor FIrpic (HOMO = −5.70 eV, LUMO = −3.10 eV), and the green-emitting phosphor Ir(ppy)_3_ (HOMO = −5.50 eV, LUMO = −2.70 eV). These phosphors exhibit a characteristic energy level alignment—featuring a shallower HOMO and a deeper LUMO—that allows their energy levels to be effectively “nested” within those of the CBP host. This configuration spatially confines the carrier recombination zone to the interior of the guest molecules, thereby enabling highly efficient electroluminescence. The phosphorescent materials employed in this study (Ir(ppy)_3_, FIrpic, Ir(pq)_2_acac) exhibit radiative decay originating from the relaxation of triplet excitons. Typically, their intrinsic radiative lifetimes (τ) are on the microsecond (µs) scale. Among them, the red-emitting dye Ir(pq)_2_acac demonstrates the longest radiative lifetime, followed by the green-emitting dye Ir(ppy)_3_, while the blue-emitting dye FIrpic possesses the shortest radiative lifetime. The 3-dB bandwidth calculation formula for the OLED device is given as follows:
(1)$$f_{-3dB}=\frac{1}{2\pi \tau }\;,$$

Due to the intrinsic radiative lifetime (
τ) of phosphorescent dyes being on the order of microseconds (µs), the 3-dB bandwidth of the OLED device is consequently limited to the kilohertz (kHz) range. Theoretically, since the blue-emitting dye FIrpic possesses the shortest radiative lifetime, it exhibits the largest 3-dB bandwidth; conversely, the red-emitting dye Ir(pq)_2_acac, with the longest radiative lifetime, demonstrates the smallest 3-dB bandwidth. The green-emitting dye Ir(ppy)_3_, with an intermediate radiative lifetime, correspondingly achieves a 3-dB bandwidth between those of the other two dyes.

In OLED-based visible light communication (OLED-VLC) systems, the selection of second-generation metal-phosphorescent complexes over third-generation (Thermally Activated Delayed Fluorescence (TADF)) or fourth-generation (Multi-Resonance Thermally Activated Delayed Fluorescence (MR-TADF)) purely organic systems is primarily based on a trade-off between device response speed and overall stability. Although TADF materials theoretically enable 100% exciton utilization efficiency, their slow reverse intersystem crossing process limits the modulation bandwidth, becoming a bottleneck for high-speed VLC. Meanwhile, MR-TADF materials, while exhibiting excellent color purity, still face challenges in efficiency and processability. In contrast, metal-phosphorescent systems achieve an optimal balance in exciton utilization efficiency, particularly in terms of higher radiative rates and short lifetimes on the microsecond or even sub-microsecond scale, which is crucial for high-speed signal modulation. Furthermore, their well-established and stable material system better satisfies the dual requirements of performance and reliability in current VLC systems.

The electron transport layer (ETL) consists of TPBi (1,3,5-Tris(1-phenyl-1H-benzimidazol-2-yl)benzene) (LUMO = −2.70 eV). The favorable energy level alignment between TPBi and the LiF/Al cathode (work function ≈ −2.90 eV) serves to lower the electron injection barrier, thereby facilitating efficient electron injection from the transport layer into the EML.

The device features a strategically graded alignment of HOMO and LUMO energy levels from the anode to the cathode [9], which systematically governs the injection, transport, and recombination dynamics of holes and electrons across the entire functional stack. The emission wavelength of the device is directly determined by the energy level characteristics of the dopants within the emissive layer. Ultimately, the synergistic optimization of the overall energy level architecture is key to realizing high-performance OLED devices.

For visible light communication (VLC) systems, key performance parameters of OLED devices—such as external quantum efficiency (EQE) and electroluminescent spectra—are of particular importance. Accordingly, the three OLED devices were characterized to evaluate their relevant performance parameters. The measured current density–voltage (J–V), luminance–current density (L–J), voltage–luminance (V–L), external quantum efficiency–current density (EQE–J), current efficiency–current density (C.E–J), and power efficiency–current density (P.E–J) characteristics are summarized in Figure 4. These curves systematically depict the electroluminescent behavior of the RGB OLEDs. In the J–V curve, the blue device was observed to show the most rapid increase in current density with voltage, which is attributed to its favorable energy level alignment. In contrast, the green device exhibited the lowest current density at the same voltage, indicating more effective charge balance. Analysis of the L–J and V–L curves jointly reveals that the green device achieved the highest luminance rise rate with increasing current density, together with the lowest driving voltage at a given luminance, underscoring its high-efficiency emission capability. The blue device, however, displayed a slower luminance increase and a significantly higher operating voltage, suggesting lower charge recombination efficiency. As observed in the EQE–J curve, all three devices initially exhibited high EQE values. A relatively gentle decline in the mid-to-high current density region (J > 25 mA/cm^2^) is observed, which is a characteristic signature of triplet-triplet annihilation (TTA) acting as the dominant mechanism. TTA is a bimolecular reaction process whose rate is proportional to the square of the exciton density. Since the exciton density is approximately proportional to the current density (J), the efficiency loss induced by TTA exhibits a J^2^ dependence. This manifests as a gradual, parabolic-like roll-off on the efficiency curve. Specifically, the moderately low doping concentrations employed in this work (Ir(ppy)_3_: 4 wt%, Ir(pq)_2_acac: 4 wt%, FIrpic: 2 wt%) were designed to effectively suppress aggregation effects among the emitting molecules. Consequently, aggregation-induced static concentration quenching is not the primary factor responsible for the observed efficiency roll-off. The phenomenon is more likely attributable to a dynamic annihilation process activated under high current densities. Given the long lifetime and high density of triplet excitons in phosphorescent materials, TTA is identified as the dominant roll-off mechanism in this regime. Simultaneously, a superimposed contribution from triplet-polaron quenching (TPQ), potentially induced by a slight imbalance in charge carrier transport, cannot be entirely ruled out. These nonlinear annihilation processes are not only key factors limiting the device efficiency but also represent the fundamental physical origin of the bandwidth limitation and signal distortion in visible light communication systems operating at high modulation frequencies. Furthermore, the C.E–J and P.E–J curves confirm that the green device maintained the highest current efficiency and power efficiency over a broad luminance range. Owing to its higher driving voltage and lower luminescent efficiency, the blue device yielded the lowest values for both metrics, while the red device showed intermediate performance.

The electroluminescent spectra of the three OLED devices were characterized using a spectrometer, as shown in Figure 5. The blue-emitting device displays a primary emission peak centered at approximately 460 nm, with a full width at half maximum (FWHM) of about 17.4 nm. The green-emitting device shows an emission peak centered around 525 nm and an FWHM of about 28.4 nm. The red-emitting device exhibits an emission peak near 620 nm, with an FWHM of approximately 33 nm.

The operational lifetime of OLED devices is a critical parameter for assessing the stability of optical communication systems [10]. In this study, the three OLED devices were evaluated for luminance decay and voltage shift at a constant current density of 30 mA/cm^2^. Figure 6 shows the normalized luminance decay curves of the red, green, and blue devices, where the horizontal axis represents the operating time
T (in hours), and the vertical axis denotes the normalized luminance
$L/L_{0}$ (%). The results indicate that the blue device exhibits the most pronounced luminance decay, followed by the green device, whereas the red device demonstrates the slowest degradation, suggesting inferior stability of the blue-emitting material compared to its red and green counterparts. It also presents the corresponding voltage shift curves, with the vertical axis indicating the voltage change
ΔV (in V). As the operating time extends, the blue device still shows the most significant voltage increase, while the voltage rise in the red and green devices remains more gradual.

Based on the aforementioned characterization, the key parameters of the three OLED devices measured at a current density of 10 mA/cm^2^ are systematically summarized in Table 1. The presented data comprehensively demonstrates the exceptional optoelectronic performance of these RGB OLED devices, thereby underscoring the significant advantages of OLED technology for visible light communication applications.

## 3. OLED Device Bandwidth Measurement System

Bandwidth serves as a key indicator for assessing the capacity of communication systems. In general, a wider bandwidth corresponds to a higher channel capacity, enabling the transmission of a greater amount of information per second. Evaluating the bandwidth of organic light-emitting diode (OLED) devices in the context of communication applications provides critical insight into their potential for visible light communication (VLC) systems. Since the bandwidth of an OLED is directly governed by its response speed, characterization of its frequency response is essential. To accurately characterize the frequency response of the OLED device and gain a deeper understanding of its modulation limits, this study establishes a small-signal equivalent circuit model, as illustrated in Figure 7. By incorporating bias-dependent physical parameters, the model reveals the intrinsic mechanisms governing the device’s bandwidth. The core of the model consists of a series resistor
$R_{s}$, and a parallel network comprising a dynamic resistor
$R_{d}$, a geometric capacitance
$C_{geo}$, and a diffusion capacitance
$C_{diff}$. Here, the series resistor
$R_{s}$, which originates primarily from the sheet resistance of the electrodes and the contact resistance, can be regarded as a bias-independent linear component. In contrast,
$R_{d}$ and
$C_{diff}$ in the parallel network are key to characterizing the nonlinear behavior of the OLED. The dynamic resistor
$R_{d}$ represents the differential resistance of the DC current-voltage (I-V) characteristic of the OLED at a specific operating point (Q). Its value, determined by the diode equation, can be approximated as:
(2)$$R_{d}={\left(\frac{\partial I_{DC}}{\partial V_{DC}}\right)}^{-1}\approx \frac{nV_{T}}{I_{DC}}\;,\;$$ where
$I_{DC}$ and
$V_{DC}$ are the DC bias current and voltage, respectively,
n is the diode ideality factor, and
$V_{T}$ is the thermal voltage.
$R_{d}$ is a nonlinear parameter with a strong dependence on the DC bias current; it decreases significantly as the current increases.

The geometric capacitance,
$C_{geo}$, originates from the parallel-plate capacitor structure of the device, formed by the top electrode, organic layers, and bottom electrode. Its value can be calculated by the following formula:
(3)$$C_{geo}=\epsilon_{0}\epsilon_{r}\frac{A}{d}\;,$$ where
A is the emitting area,
d is the total thickness of the organic layers,
$\epsilon_{0}$ is the vacuum permittivity with a value of approximately
$8.854\times 10^{-12}\;F/m$, and
$\epsilon_{r}$ is the average relative permittivity of the organic materials.
$C_{geo}$ is a bias-independent constant.

The diffusion capacitance,
$C_{diff}$, is the most critical capacitor in the small-signal equivalent circuit model of an OLED device. It characterizes the charge storage effect resulting from carriers accumulating in the active region for recombination. Its value is approximately proportional to the carrier transit time
$\tau_{c}$ and inversely proportional to the dynamic resistance
$R_{d}$, and can therefore be expressed as:
(4)$$C_{diff}\approx \frac{\tau_{c}}{R_{d}}\;,$$

$C_{diff}$ is likewise a nonlinear parameter and exhibits a strong dependence on the DC bias. Under normal operating conditions, the value of
$C_{diff}$ is typically significantly larger than that of
$C_{geo}$, thereby becoming the dominant component of the device’s total capacitance. Therefore, the total equivalent capacitance of the device,
$C_{total}$, is expressed as:
(5)$$C_{total}=C_{geo}+C_{diff}\;,$$

Based on the model shown in Figure 7, the frequency response of the device can be derived. The total impedance,
$Z_{total}(\omega )$, equals the sum of the series resistance and the impedance of the parallel section,
$Z_{p}(\omega )$, and is given by:
(6)$$Z_{total}\left(\omega \right)=R_{s}+Z_{p}\left(\omega \right)=R_{s}+\frac{R_{d}}{1+j\omega R_{d}C_{total}}\;,$$

When the input is a voltage signal
$V_{in}$, the output signal
$V_{out}$ is taken across the parallel network. The voltage transfer function
H(ω) is given by:
(7)$$\;H\left(\omega \right)=\frac{V_{out}}{V_{in}}=\frac{Z_{p}\left(\omega \right)}{Z_{total}\left(\omega \right)}=\frac{\frac{R_{d}}{1+j\omega R_{d}C_{total}}}{R_{s}+\frac{R_{d}}{1+j\omega R_{d}C_{total}}}\;,$$

By simplifying this expression, it can be reduced to the standard form of a first-order low-pass filter:
(8)$$H\left(\omega \right)=\frac{V_{out}}{V_{in}}=\frac{R_{d}}{R_{s}+R_{d}}\cdot \frac{1}{1+j\omega \frac{R_{s}R_{d}}{R_{s}+R_{d}}\cdot C_{total}},$$

From the above equation, it follows that the system time constant
τ is given by:
(9)$$\tau =\frac{R_{s}R_{d}}{R_{s}+R_{d}}\cdot C_{total}\;,$$

Correspondingly, the system’s 3-dB bandwidth (
$f_{-3\mathrm{dB}}$), defined as the cutoff frequency where the frequency response drops to −3 dB, is given by:
(10)$$f_{-3dB}=\frac{1}{2\pi \tau }=\frac{1}{2\pi \cdot \frac{R_{s}R_{d}}{R_{s}+R_{d}}\cdot C_{total}}=\frac{R_{s}+R_{d}}{2\pi R_{s}R_{d}C_{total}}\;,$$

This expression clearly reveals the intrinsic mechanism governing the modulation bandwidth: the bandwidth is not determined by a single parameter, but is governed by the combined effect of the series resistance
$R_{s}$, the dynamic resistance
$R_{d}$, and the total capacitance
$C_{total}$.

Under normal operating bias, the condition
$C_{diff}\gg C_{geo}$ typically holds. It follows that the modulation bandwidth of the device is ultimately limited by the diffusion capacitance
$C_{diff}$, whose physical origin lies in the finite carrier mobility and recombination lifetime within the organic semiconductor material. The strong nonlinear dependence of both
$R_{d}$ and
$C_{diff}$ (and hence
$C_{total}$) on the DC bias current
$I_{DC}$ is the fundamental reason for the observed variation in bandwidth with driving conditions. Consequently, the essential approach to enhancing the OLED modulation bandwidth lies in developing novel organic materials with higher carrier mobility, which reduces the carrier transit time
$\tau_{c}$ and thereby lowers
$C_{diff}$.

Given that OLEDs typically exhibit limited bandwidth (on the order of kHz), a photodetector with a substantially higher bandwidth—such as the DET36A2 (DP), with approximately 25 MHz—was used as the receiver to measure the frequency response. Under this configuration, the overall link bandwidth can be largely attributed to that of the OLED device [11].

The physical setup of the bandwidth measurement system is illustrated in Figure 8. The OLED device was operated under linear bias conditions, with a swept-frequency signal (1 Hz to 2 MHz) generated by an arbitrary function generator (GW Instek AFG-2230, GW Instek, Suzhou, China) at a peak-to-peak voltage (
$V_{pp}$) of 100 mV. This signal was coupled to the OLED device through a Bias-tee (ZFBT-6GW-FT+, Mini-Circuits, Brooklyn, NY, USA), which serves as a hybrid circuit combining DC bias and AC modulation. As shown in Figure 9, the optical path incorporates two plano-convex lenses for light transmission. The first lens (flat surface facing the OLED device, convex surface facing outward) collimates the divergent emission from the OLED device to reduce optical loss. The second lens, similarly oriented, focuses the collimated beam onto the photodetector. At the receiver, the time-domain signal captured by the photodetector was converted to the frequency domain using the spectrum analysis function of an oscilloscope (GW Instek MDO-2204ES, GW Instek, Suzhou, China), and the frequency response was recorded. As shown in Figure 10, the measured 3-dB bandwidth under normal operating conditions is approximately 466 kHz for the blue OLED device, 308 kHz for the green OLED device, and 284 kHz for the red OLED device.

## 4. Visible Light Communication System Simulation

In organic light-emitting diode (OLED)-based visible light communication (OLED-VLC), data is transmitted by modulating information onto light beams that propagate through free space via point-to-point transmission. This method not only avoids electromagnetic pollution but also ensures no radiation harm to the human body. The architecture of OLED-VLC shares a structural similarity with optical fiber communication, consisting primarily of an optical transmitter module and an optical receiver module [12,13], as depicted in Figure 11. In the transmitter module, processed binary bitstreams are used to drive the OLED light source. Intensity modulation is applied to the OLED, converting electrical signals into optical signals that are emitted into free space. Since the human eye cannot perceive such high-frequency intensity modulation, the system simultaneously fulfills basic illumination requirements in the environment [14]. The optical signal carrying the data propagates through free space and is collected by an avalanche photodiode (APD) at the receiver module. The APD converts the incident optical signal into an electrical signal, which is subsequently amplified, demodulated, and filtered to recover the original transmitted data.

On-off keying (OOK) modulation is widely adopted in visible light communication (VLC) systems [15], where a binary “1” is represented by the presence of a carrier signal over a specific duration, while a binary “0” is indicated by its absence. OOK modulation schemes are primarily categorized into non-return-to-zero (NRZ-OOK) and return-to-zero (RZ-OOK) formats [16]. In this work, the NRZ-OOK scheme was implemented. Accordingly, leveraging principles of visible light communication and wavelength division multiplexing (WDM), a free-space VLC system based on RGB OLEDs signal sources was designed by modeling OLED characteristics in simulation software. The simulated system architecture is presented in Figure 12.

At the transmitter, a random bit sequence was generated and fed into a non-return-to-zero (NRZ) pulse generator. The resulting electrical pulses were used to directly intensity-modulate three independent light sources. The modulated optical signals at different wavelengths were then combined using a WDM and coupled into a free-space optical (FSO) channel. Since each signal is carried by a distinct wavelength, they propagate through the free space without mutual interference [17,18,19]. At the receiver, a demultiplexer separates the combined beam. To distinguish signals from different wavelengths, an optical bandpass filter was positioned prior to the APD photodetector to eliminate stray light carrying other information, thereby effectively suppressing background optical interference. The detected electrical signal was then processed through a low-pass Bessel filter to remove out-of-band noise and recover the original transmitted data. To evaluate the transmission performance of the system, optoelectronic instruments including an optical spectrum analyzer, oscilloscope, and bit error rate tester were utilized.

The simulation parameters for the designed OLED-based visible light communication system were configured as follows. Based on the previously characterized parameters of the three OLED devices, the red OLED wavelength was set at 620 nm. Accordingly, the transmission range of the corresponding optical bandpass filter was configured from 610 nm to 630 nm. With an external quantum efficiency of 37.9% and a 3-dB bandwidth of 284 kHz, the cutoff frequency of the corresponding low-pass Bessel filter was set to 300 kHz. The green OLED wavelength was set at 525 nm, and its corresponding optical bandpass filter transmission range was set from 515 nm to 535 nm. With an external quantum efficiency of 39.9% and a 3-dB bandwidth of 308 kHz, the cutoff frequency of the corresponding low-pass Bessel filter was set to 400 kHz. The blue OLED wavelength was set at 460 nm, and its corresponding optical bandpass filter transmission range was set from 450 nm to 470 nm. With an external quantum efficiency of 21.1% and a 3-dB bandwidth of 466 kHz, the cutoff frequency of the corresponding low-pass Bessel filter was set to 500 kHz. The pseudorandom bit sequence (PRBS) length was set to 256 bits—a balance that was chosen to ensure statistical reliability while maximizing simulation efficiency. This trade-off aligns well with the practical requirements of low power consumption and low complexity in flexible displays and wearable devices. The FSO channel attenuation was set to 2.5 dB/km, simulating signal attenuation under clear indoor atmospheric conditions with good visibility and absence of heavy fog, haze, or smoke, where the primary loss stems from geometric loss. The transmission distance was set to 2 m, accurately reflecting the typical interaction range in personal area networks for wearable devices and flexible displays. The transmitter beam divergence was set to 2816 mrad, precisely reproducing the omnidirectional radiation characteristics of an OLED area source. The optical transmitter aperture of the signal source was 25 mm, simulating the dimensions of the OLED device supplied by PhiChem Corporation. The APD photodetector parameters were set as follows: a receiving aperture of 10 mm, a bandwidth of 5 MHz, a dark current of 10 nA, a responsivity of 1 A/W, and a gain of 50. The 10 mm receiving aperture met the miniaturization requirements of wearable devices; the 5 MHz bandwidth fully covered the maximum OLED bandwidth with sufficient margin to ensure distortion-free signal transmission; the gain of 50 was a key optimization, effectively amplifying the weak signals resulting from the OLED’s large divergence angle while avoiding excessive noise amplification; the dark current of 10 nA and the responsivity of 1 A/W represented typical values for commercial APD photodetectors.

The three RGB signal channels were combined via a multiplexer and subsequently transmitted through the free-space wireless channel. As shown in the optical spectrum analyzer results (Figure 13), the resulting spectrum exhibits clearly separated emission peaks at their respective wavelengths. This wavelength-division configuration not only enhanced the overall channel capacity but also alleviated bandwidth constraints [20]. Moreover, the blended white light generated by this approach demonstrated a high color rendering index (CRI), thereby fulfilling the illumination requirements in practical VLC scenarios.

The signal was transmitted through the FSO channel via the WDM. The output signal was detected and captured at the receiver using an oscilloscope. As shown in Figure 14, a comparison between the original and received signals demonstrates that the waveform of the received signal closely matches that of the original. The overall waveform distortion was found to be minimal. After processing, the received signal satisfactorily met the required communication standards.

The transmission signal quality was further evaluated using the maximum Q-factor and minimum bit error rate (BER) obtained from a BER analyzer, which provide direct indicators of signal integrity. Higher Q-factor values and lower BER correspond to superior signal quality. The resulting eye diagrams are presented in Figure 15. As observed, all three-color channels exhibit clear eye diagrams with wide openings, indicating high signal quality. The signals show relatively fast rise and fall times, and the optimal receiver sensitivity is achieved when the crossing point ratio is approximately 45%. Under identical conditions, the eye-opening dimensions of the three colors are comparatively similar. The red channel achieved a maximum Q-factor of 10.79 with a minimum BER of 1.67 × 10^−27^, the green channel reached a maximum Q-factor of 12.30 with a minimum BER of 3.74 × 10^−35^, and the blue channel yielded a maximum Q-factor of 8.98 with a minimum BER of 1.18 × 10^−19^. All bit errors can be successfully identified and corrected when the BER remains below the forward error correction (FEC) threshold. Thus, a BER below the typical FEC limit of 3.8 × 10^−3^ is considered acceptable for reliable communication. In this system, the measured BER values are well below the FEC threshold, confirming robust and smooth data transmission. These results support the potential application of OLED-based systems in emerging fields such as flexible displays and wearable devices, where secure and reliable optical communication is essential.

## 5. Conclusions

This study systematically characterized the key optoelectronic properties of red, green, and blue (RGB) organic light-emitting diode (OLED) devices, with precise measurement of core parameters such as modulation bandwidth that critically determine communication performance. Based on the experimental characterization, a simulation model of a visible light communication (VLC) system incorporating wavelength division multiplexing (WDM) was successfully constructed. The results demonstrate that the blue OLED device exhibits the highest modulation bandwidth (approximately 466 kHz), followed by the green and red devices. All three devices show stable emission spectra and well-defined chromaticity coordinates, forming a solid physical basis for WDM implementation. System simulation under Non-Return-to-Zero On-Off Keying (NRZ-OOK) modulation reveals excellent transmission performance, with all three channels achieving Q-factors above 8.98 and bit error rates lower than 10^−19^. These findings confirm the feasibility of OLEDs in supporting multi-channel parallel transmission at moderate data rates, and further highlight their unique potential for the integration of display, communication, and illumination functions. This work provides essential data support and systematic validation for translating OLED-based VLC technology from theoretical concept to practical application, thereby offering a promising technical pathway to address the growing demand for highly integrated, low-power communication in emerging 6G scenarios such as flexible terminals and wearable devices.

## Figures and Tables

**Figure 1 micromachines-16-01338-f001:**
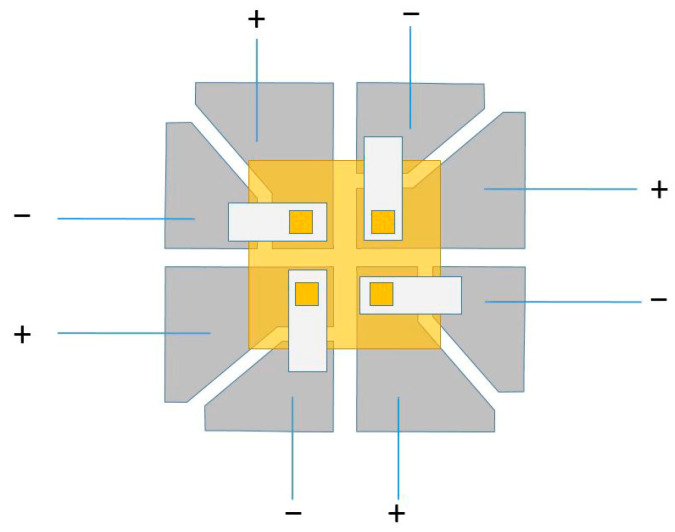
OLED device structure.

**Figure 2 micromachines-16-01338-f002:**
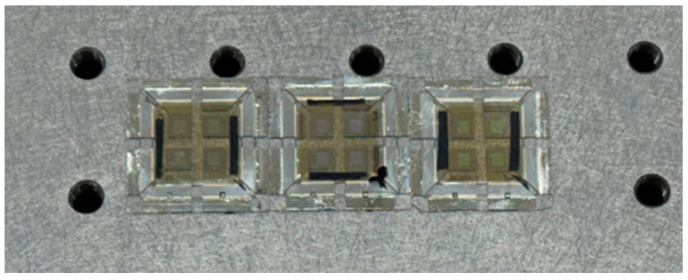
RGB OLED devices.

**Figure 3 micromachines-16-01338-f003:**
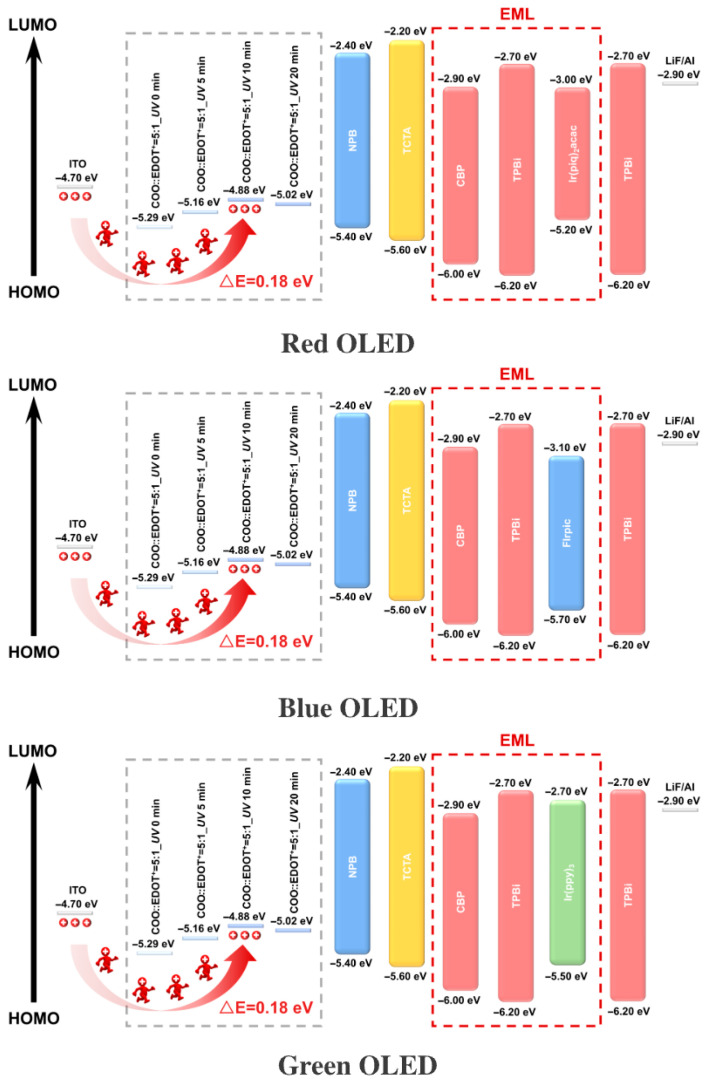
Energy level of OLED devices.

**Figure 4 micromachines-16-01338-f004:**
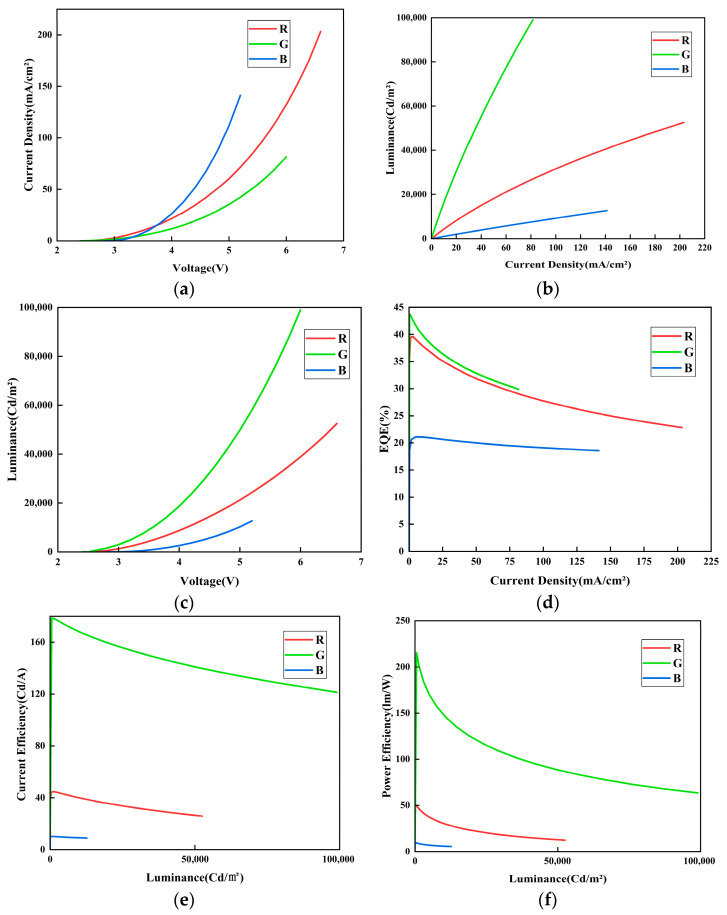
(**a**) J-V Curve. (**b**) L-J Curve. (**c**) L-V Curve. (**d**) EQE-J Curve. (**e**) C.E-L Curve. (**f**) P.E-L Curve.

**Figure 5 micromachines-16-01338-f005:**
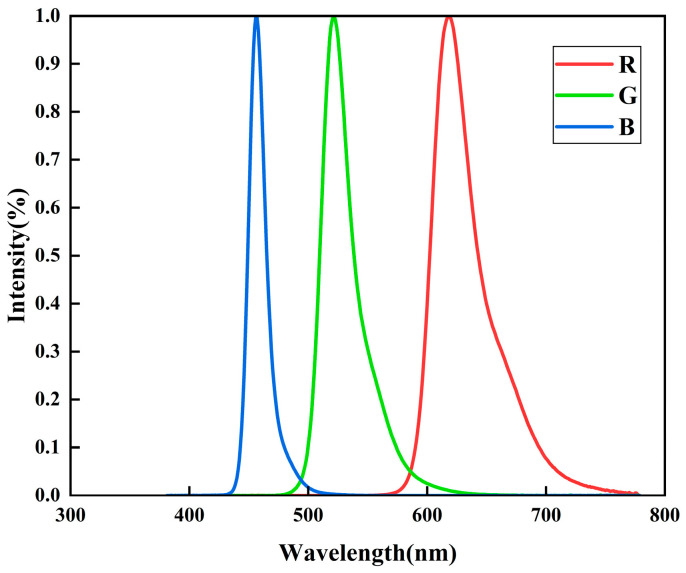
EL spectra of OLED devices.

**Figure 6 micromachines-16-01338-f006:**
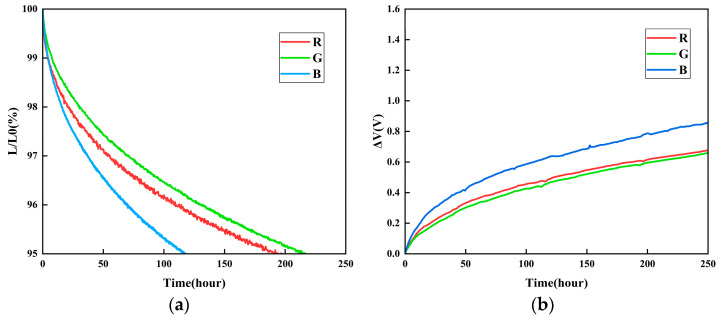
(**a**) Luminance decay curve. (**b**) Voltage decay curve.

**Figure 7 micromachines-16-01338-f007:**
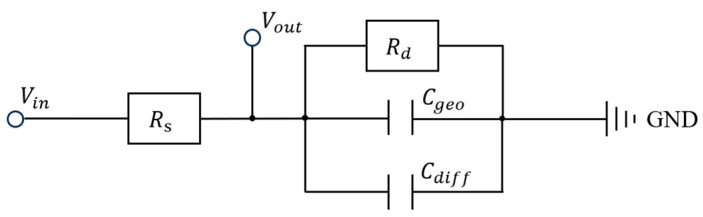
OLED device small-signal equivalent circuit model.

**Figure 8 micromachines-16-01338-f008:**
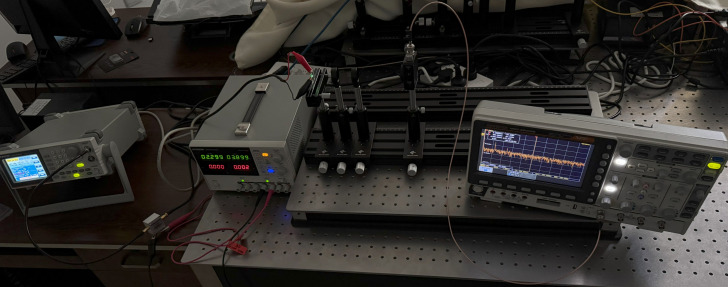
Photo of the OLED device bandwidth measurement system.

**Figure 9 micromachines-16-01338-f009:**
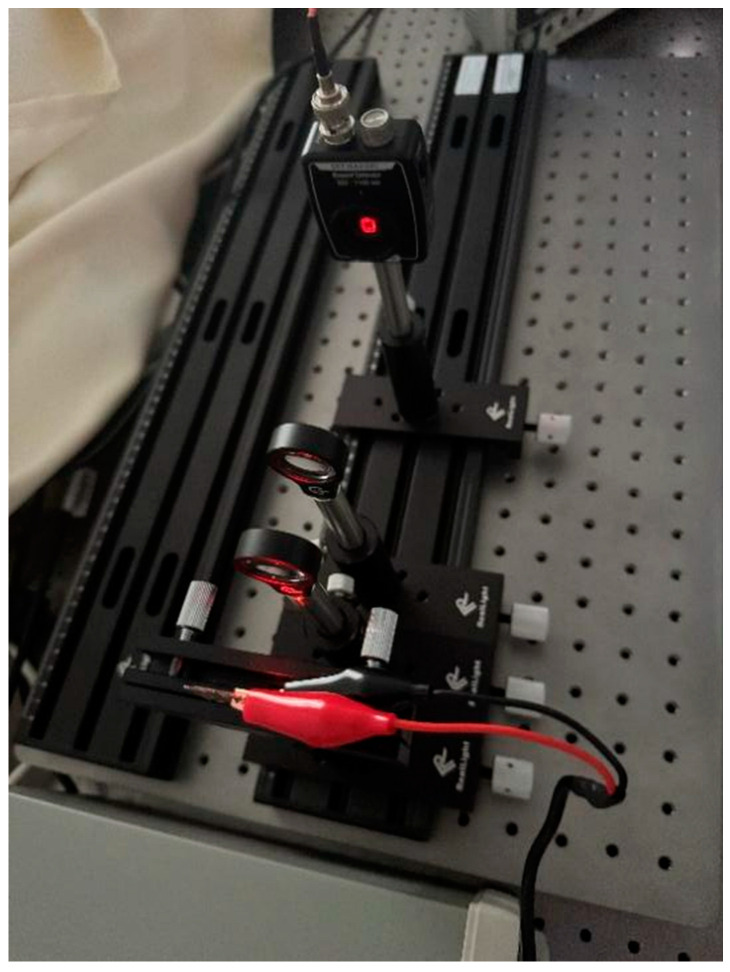
Signal transmission interface.

**Figure 10 micromachines-16-01338-f010:**
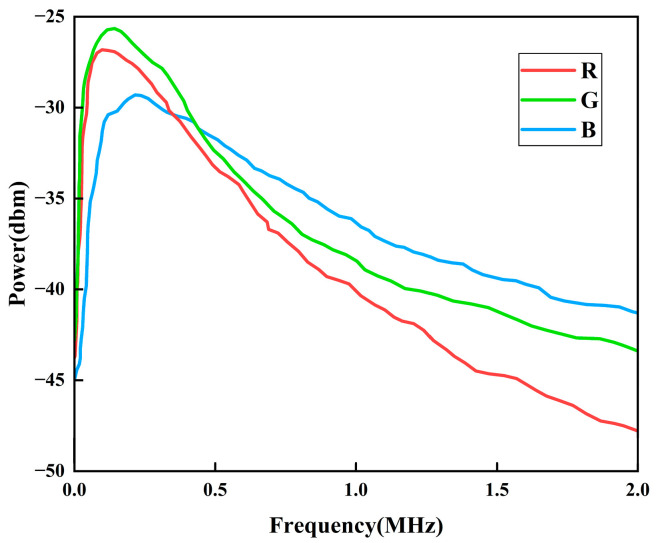
Frequency response curve of OLED devices.

**Figure 11 micromachines-16-01338-f011:**
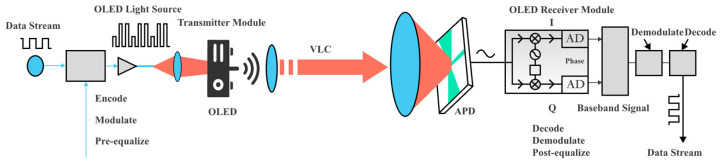
Schematic of the OLED-based VLC system.

**Figure 12 micromachines-16-01338-f012:**
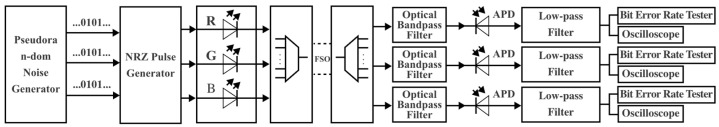
Simulation of the VLC system.

**Figure 13 micromachines-16-01338-f013:**
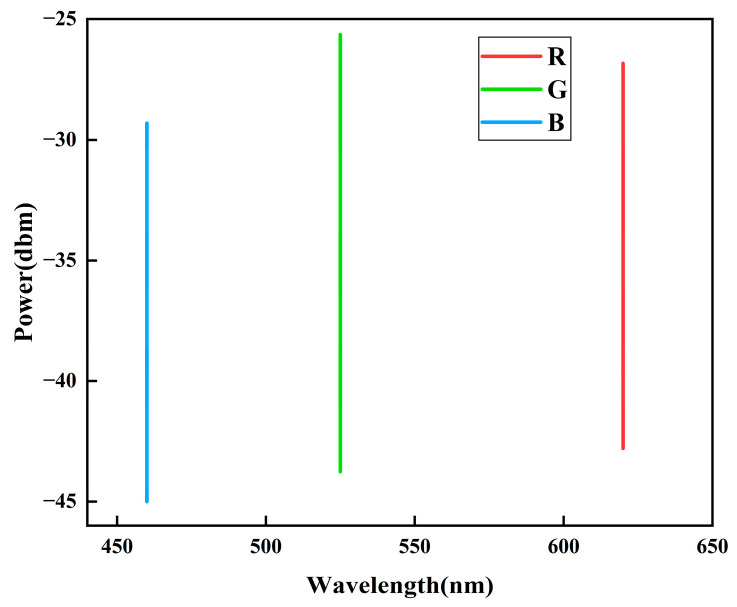
Power spectrums of RGB channels.

**Figure 14 micromachines-16-01338-f014:**
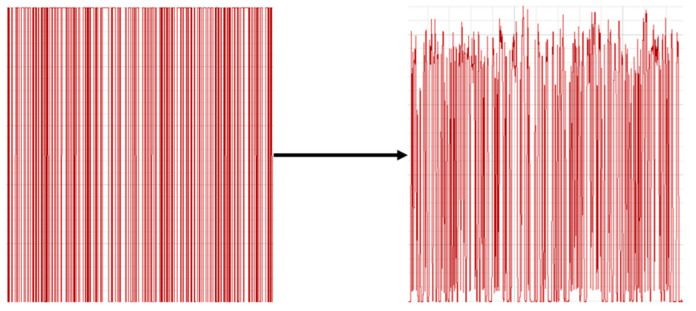
Transmitted signal and received signal.

**Figure 15 micromachines-16-01338-f015:**
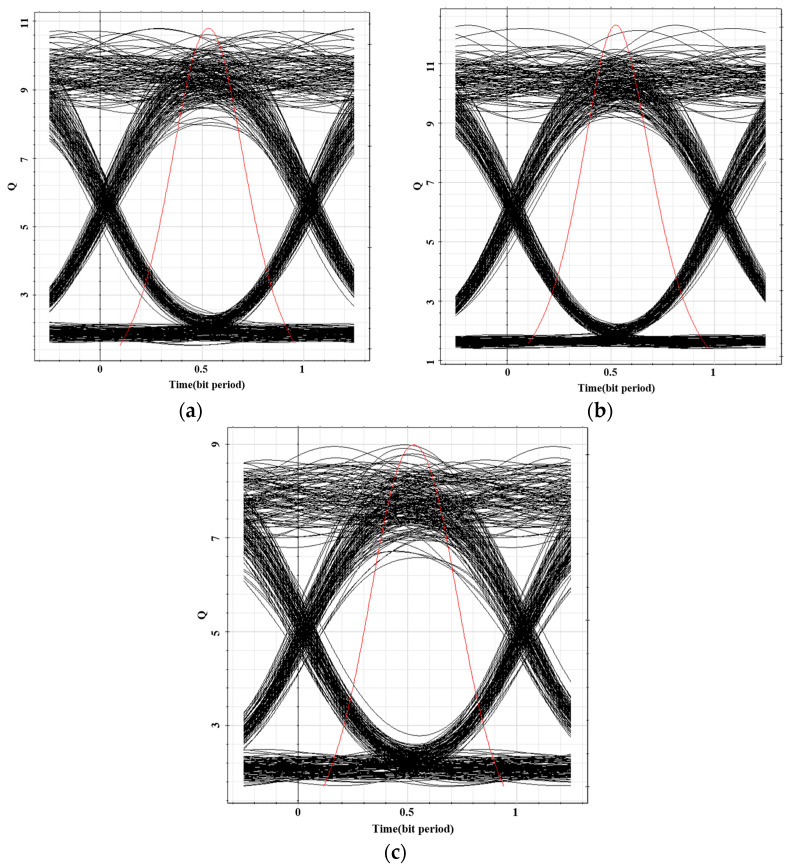
(**a**) Eye diagram showing the maximum Q-factor for Channel R. (**b**) Eye diagram showing the maximum Q-factor for Channel G. (**c**) Eye diagram showing the maximum Q-factor for Channel B.

**Table 1 micromachines-16-01338-t001:** The optoelectronic characteristics of the OLED devices at a current density of 10 mA/cm^2^.

J = 10 mA/cm^2^	L	V	C.E.	EQE	P.E.	CIEx	CIEy	EL	FWHM
R	4298	3.49	43.06	37.9	38.8	0.682	0.318	620	33.0
G	16,218	3.88	162.43	39.9	131.5	0.231	0.720	525	28.4
B	1018	3.59	10.18	21.1	8.9	0.138	0.046	460	17.4

## Data Availability

The original contributions presented in this study are included in the article. Further inquiries can be directed to the corresponding author.

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
