# Peer review of "A Study on Visible Light Communication Systems Based on OLED Devices"

_micromachines, 2025, doi:10.3390/mi16121338_

Round 1
Reviewer 1 Report
Comments and Suggestions for Authors
I’ve thoroughly reviewed the manuscript titled “A Study on Visible Light Communication System Based on OLED Devices.” While the topic is relevant and the experimental OLED characterization is valuable, the title suggests that this is a paper about a communication system rather than an OLED device paper. Consequently, the manuscript requires improvements before it can be considered for publication. Below is a list of issues that should be addressed.
- Lack of a small-signal OLED model. The paper presents measured bandwidths but does not include a small-signal equivalent circuit model (RC model, parasitic capacitance, impedance characteristics). A proper small-signal model is essential for understanding modulation limits and validating the communication analysis.
- Missing large-signal modulation behavior. OLEDs are highly nonlinear devices. The manuscript only analyzes linear small-signal response, with no investigation of large-signal effects such as waveform distortion, turn-on delay, slew-rate limitations, or BER degradation under higher modulation depth.
- Simulation parameters not fully justified. Several parameters (APD responsivity, filter cutoff frequency, PRBS length, channel attenuation, noise model) are included without explanation or reference. These choices should be justified or validated.
Reviewer 2 Report
Comments and Suggestions for Authors
Please find it attached.

Round 2
Reviewer 1 Report
Comments and Suggestions for Authors
The authors have addressed my concerns, and I suggest publishing.